# Technology and Process Design for Phenols Recovery from Industrial Chicory (*Chicorium intybus*) Leftovers

**DOI:** 10.3390/molecules24152681

**Published:** 2019-07-24

**Authors:** Camilla Maria Cova, Luisa Boffa, Marco Pistocchi, Silver Giorgini, Rafael Luque, Giancarlo Cravotto

**Affiliations:** 1Departamento de Química Orgánica, Universidad de Córdoba, Ctra Nnal IV, 14014 Córdoba, Spain; 2Dipartimento di Scienza e Tecnologia del Farmaco, University of Turin, Via P. Giuria 9, 10125 Turin, Italy; 3BioSphere Srl, Via Cellaimo, 3456, 47032 Bertinoro (FC), Italy; 4Orogel SpA, Via Dismano 2600, 47522 Cesena (FC), Italy

**Keywords:** chicory leftovers, ultrasound-assisted extraction, microwave-assisted extraction, simultaneous ultrasound/microwave extraction, subcritical water, total polyphenolic content

## Abstract

Vegetal leftovers from the agro–food industry represent a huge source of primary and secondary metabolites, vitamin, mineral salts and soluble as well as insoluble fibers. Economic reports on the growth in the polyphenol market have driven us to focus our investigation on chicory (*Chicorium intybus* L.), which is one of the most popular horticultural plants in the world and a rich source of phenolic compounds. Ultrasound-assisted extraction (UAE), microwave-assisted extraction (MAE) and their simultaneous combination, using either ethanol/water or water alone (also sub-critical), have been investigated with the aim of designing a green and efficient extraction process. Higher total-polyphenol yields as well as dramatic reductions in extraction times and solvent consumption have been obtained under these conditions. ANOVA test for analyses of variance followed by the Tukey honestly significant difference (HSD) post-hoc test of multiple comparisons was used in the statistical analysis. MAE experiments performed with sub-critical water, and MW/US experiments with an ethanol solution have shown polyphenol recovery values of up to ~3 g of gallic acid equivalents (GAE) per kg of fresh material in only 15 min, while conventional extraction required 240 min to obtain the same result.

## 1. Introduction

The key to the valorisation of agro–food industrial leftovers currently lies in the development of convenient strategies and technologies that can address economic and environmental demands. These new technologies should provide process-intensification and energy-saving characteristics, which entail the design of more efficient extraction methods that make use of non-conventional energy sources. In addition, these new processes should fulfil the needs for greener and more sustainable protocols [1], as expressed by the food and cosmetic industries. With these aims in mind, green extraction techniques have been developed as valid alternatives to traditional methods; they shorten processing times and decrease energy consumption. These methods also permit the amount of consumed solvents to be reduced and promote the exploitation of safer and less toxic solvents [2]. Furthermore, non-traditional techniques can defend the extract from thermal degradation, thus preserving its target activities [3,4,5] Examples of green, efficient extraction technologies include ultrasound-assisted extraction (UAE), microwave-assisted extraction (MAE) and the combination of these techniques. 

Ultrasound (US, 18–40 kHz) processes can be significantly less time consuming and energy intensive. When applied to extraction methods, in so-called UAE, US also protects the extract from thermal degradation phenomena thanks to its low bulk temperatures and fast process times. In addition, no mechanical parts are moved inside the extract when using US and no residues are left. Finally, cavitation phenomena mean that US can give higher extraction efficiencies [6]. 

MAE on the other hand, can be carried out in very short times and provides good reproducibility. Moreover, the MAE technique allows solvent volumes to be reduced, making manipulation easier and leading to higher final-extract purity using only a small fraction of the energy that traditional methods normally require. The volumetric heating produced by microwaves (MW) has numerous benefits, which are delivered by quicker energy transfer, reduced thermal gradients and exceptional heating selectivity [1]. 

The exploitation of MW and US technologies simultaneously in a single device allows efficient heat transfer (MW) to be combined with the cavitation phenomenon and mass transport (US). This effect has led to enhancements in the efficiency of extraction processes [7]. In addition, the non-conventional extraction techniques, UAE and MAE, can be applied on pilot and industrial scales and can provide worthwhile gains in extraction efficiency and economy.

Polyphenols have received notable consideration in recent years because of their extraordinary health benefits, such as possibly reducing the probability of contracting chronic diseases. This class of compounds also possesses remarkable antioxidant capacity, radical-scavenging activity and the ability to chelate transition metal ions [8,9]. Furthermore, polyphenols should always be consumed daily, since the human body is unable to synthesise them [10]. As a consequence, the extraction of polyphenols from plants is of primary importance for the production of additives for the food and cosmetic industries [11].

Chicory (*Chicorium intybus* L.), a perennial herbaceous plant belonging to the Asteraceae family, is interesting and less frequently studied than the majority of plants under research for polyphenol extraction. Its leaves have a range of uses, including as ingredients in salads, as fodder plants, feed additives or as herbs for the preparation of decoctions and infusions. Its roots are normally used for the preparation of chewing gum [12]. Chicory is defined as a versatile medicinal plant as it possess antidiabetic, antitoxic, antiulcerogenic, anticarcinogenic and anti-inflammatory properties [13,14,15,16,17,18]. Remarkably, these properties have been known for centuries, even the ancient Romans and Greeks used chicory as a medicine [19]. The health benefits of chicory originate from its peculiar phytochemical composition. In fact, chicory has high contents of flavonoids, anthocyanins, cinnamic and quinic acids [13,14,15,16,17,18,19,20]. The low cost of chicory and its high polyphenol content mean that the extraction of bioactive compounds from this plant is gaining attention day by day.

UAE, MAE and their combination (MW/US assisted extraction) are efficient, unconventional techniques that can be used for chicory treatment. Most investigations into *C. intybus* have dealt with inulin extraction from roots [21,22], and only a few papers have characterised leaf and/or root extracts [13,14,15,16,17,18,19,20,21,22,23]. However, to the best of our knowledge, only a few of these studies have proposed the use of MAE or UAE of phenolics [24,25,26] and there are no studies on the combination (MW/US) of these techniques for the recovery of polyphenols from chicory.

Pradal et al. have recently investigated a range of US parameters for polyphenol extraction from ground chicory, with ~1.5 g gallic acid equivalents (GAE) per 100 g d. w. being the best result [25]. This outcome was obtained using EtOH 60% at 60 °C and 100 W for 120 min. 800 mg GAE/100 g d.w has been recovered under the same conditions, but after a reduced operating time of 15 min. Moreover, Baiano et al. have reported the MAE and conventional extractions of antioxidants from vegetable solid wastes, including chicory leaves and stems [24]. The best data obtained using MAE was ~400 mg GAE/kg of fresh material with the experiment being performed at 80 °C for 4 min.

Herein, the best values in terms of phenols recovery were achieved under MAE using sub-critical H_2_O and using MW/US and EtOH 60% solution. 

## 2. Results and Discussion

### 2.1. Water Content Determination

Water content of frozen chicory leftovers was 92.8 ± 0.29 and 92.9 ± 0.17 weight/weight (*w/w*) percentage, obtained respectively from the freeze-drying and hot drying methods, while the combined value was 92.9 ± 0.22% *w/w*.

### 2.2. Extraction Conditions

The aim of this work was to investigate the influence of UAE, MAE and their combination on the rapid and selective recovery of the polyphenolic fraction from *C. intybus* leftovers. The efficacy of these unconventional extractions underwent a preliminary evaluation, which was compared with the traditional and exhaustive results. 

The choice of extraction conditions and parameters such as solvent, solid/liquid ratio, time and temperature was fundamental. 

As regards the solvent choice, the most used solvents for extraction from vegetal matrix are H_2_O, MeOH, EtOH or their mixtures [27,28,29]. Taking into account the possible use of chicory extracts into food industry, only solvents suitable for food applications, such as EtOH and H_2_O, should be used. Rosello–Soto et al. showed that H_2_O and hydroalcoholic mixtures are efficient solvent in UAE [30]. In addition, EtOH is a good microwave absorber (ε = 25.7), resulting in a good choice also for MAE [31]. Moreover, in a recent work [25], it has been demonstrated that for EtOH mixtures over 60% *v/v*, the extraction yield and the polyphenols recovery decreased and pure EtOH seemed to be ineffective in chicory wastes. For all the above-mentioned reasons, pure H_2_O and a mixture of EtOH 60% *v/v* were selected as solvents for UAE, MW/US and conventional extraction. As far as MAE is concerned, only H_2_O under subcritical conditions was tested (MW–sbc–H_2_O in the Table 1). Subcritical H_2_O extraction is usually performed using hot H_2_O (from 100 to 374 °C) under high pressure (from 10 to 60 bar) to maintain water in the liquid state. In the present work, 150 °C and 20 bar N_2_ were selected as extraction parameters.

In many applications, a solid/solvent ratio from 1:10 (g/mL) to 1:20 (g/mL) was found to be ideal both for UAE and MAE [31,32,33]. A solvent/liquid ratio 1:15 (g/mL) was selected for all the experiment to completely immerse the matrix.

During MAE, UAE and MW/US extractions, the extraction time is a key factor. Over-exposure to MW radiation or sonication for longer times leads to a decrease of the extract yield and polyphenol recovery due to the degradation of chemically active principles present in plant matrices such as phenolic compounds. To avoid thermal degradation of these compounds, the exposure times for MAE and UAE processes usually ranged from a few minutes to 30 min [34]. In this study, an extraction time of only 15 min was selected, based on our previous experience, in order to shorten the process.

Finally, different temperatures were selected according to the process. As explained before, it is generally accepted that polyphenols can suffer from thermal degradation, meaning that extended exposure to high temperatures should be avoided [5]. Concerning UAE, in order to circumvent thermal degradation and to keep an efficient US cavitation (which is contrasted by the solvent boiling bubbles), extraction temperatures normally used reach at maximum 50 °C. In our work, 40 °C was selected for UAE and a conventional maceration was carried out at the same temperature and time in order to make a comparison (US and M40 in the Table 1). For the simultaneous MW/US assisted processes, the reached temperature is usually higher than UAE because of the MW heating contribution. A temperature of 75 °C was tested and a conventional extraction was performed at the same temperature and time (MW/US and M75 in the Table 1).

Summing up, the solid/solvent ratio (1 g of matrix/15 mL of solvent) and extraction time (15 min) were kept constant across all the extractions using pure H_2_O or EtOH 60% *v/v* as solvent, while an exhaustive extraction was performed using EtOH 75% *v/v* at reflux for 240 min in order to reach the maximum yield (EM in the Table 1).

### 2.3. Extraction Yield and Phenolic Content

Extraction efficiency was evaluated across a series of single-extraction steps (UAE, MAE and MW/US) that were carried out on the same frozen matrix (H_2_O content ~93% *w/w*, see Section 2.1). Table 1 compares the obtained yields, in terms of percentage of dried extract (DE) weight over dried matrix (DM) weight, for each sample. The analyses were performed in triplicate and the results are reported with their standard deviations (SD) in Table 1.

As shown in Table 1, extraction yields ranged from 34% to 96%. The 96% extraction yield was obtained using the exhaustive method (4 h under reflux at 85 °C). MAE with sub-critical H_2_O and MW/US with H_2_O gave yields near 65%, whereas the MW/US technique in an EtOH 60% *v/v* solution for 15 min afforded the highest extraction yield (87%). As expected, extractions in EtOH 60% solutions gave higher yields than those carried out in pure H_2_O. Moreover, UAE and MW/US extractions gave higher yield compared to the conventional ones performed at the same temperature. A comparison of UAE, MAE and MW/US shows that the efficiency increased in the following order: UAE < MAE < MW/US. 

The ANOVA test determined a statistically significant difference between groups (see Section 3.7). Tukey HSD test through the comparison of all pairs of means showed the honestly significant differences between extraction protocols. As shown in Table 1, no significant differences were found between yields obtained in M40-EtOH 60% and US-H_2_O (b and c), US-H_2_O and M75-H_2_O (c and e), MW/US-H_2_O and MW-sbc-H_2_O (g and j). UAE allows the use of H_2_O instead of EtOH at lower temperature, also when combined to MW. 

The total phenolic content (TPC) of the extracts obtained under different extraction conditions is depicted in Table 1, which were selected according to our previous studies of the UAE of polyphenols [35]. Results are illustrated as TPC, which is expressed as gallic acid equivalents (GAE, mg/g) over dried extract (DE) weight, GAE (mg/g) over DM weight and GAE (g/kg) over frozen matrix (FM), for each sample. The analyses were performed in triplicate and the results are reported with their standard deviations (SD) in Table 1. 

The highest phenolic content over dried extract (67.5 mg GAE/g DE) was obtained using MAE in subcritical H_2_O for 15 min. The result obtained using MW/US and EtOH 60% *v/v* was also remarkable (49.7 mg GAE/g DE). Unconventional methods (UAE, MAE and the combination of US and MW) gave higher phenolic contents than the traditional methods. 

In terms of phenolic content over the dried matrix, 45 mg GAE/g DM were obtained from exhaustive extraction in 240 min. This value was the same obtained in only 15 min by MAE and sub-critical H_2_O, and by the combined MW/US protocol with a 60% *v/v* EtOH solution. Extraction time was therefore reduced by near 94%. In general, the TPC values obtained using the EtOH solutions were significantly higher than those obtained using H_2_O, except for MW-sbc-H_2_O sample. A comparison of UAE, MAE and MW/US revealed that efficiency increased in the following order: UAE < MW/US < MAE. 

In Table 1, it can be seen that TPC values over a kilogram of frozen matrix (FM) ranged from 0.7 to 3.1 g GAE/1 kg of frozen chicory.

Baiano et al. have recently reported the MAE of polyphenols from chicory leaves and stems [24]. The best data obtained using MAE (80 °C for 4 min) was ~0.4 g GAE/kg of fresh material, a value that was approximately 1/7 of one obtained with MAE with subcritical water in this work (~3.1 g).

A comparison of the results of the conventional method and those of the unconventional extraction techniques underlines the advantages that US-, MW- and MW/US-assisted processes provide, in terms of time required, selectivity and sustainability, which are due to reduced solvent consumption. 

In order to obtain extracts that were richer in the polyphenolic fraction, MW-sbc-H_2_O and MW/US-EtOH 60% extracts were purified using solid phase extraction (SPE) (see Section 3.5), as they were the extracts that showed the highest TPC values. After purification, the extracts were analysed again using the Folin–Ciocalteau test to give the new TPC values. The analyses were performed in triplicate and TPCs of the extracts after purification SPE with their standard deviations (SD) are depicted in Table 1. Results are illustrated as TPC, expressed as gallic acid equivalents (GAE, mg/g) over dried extract (DE) weight. A comparison of TPC values obtained before and after SPE (see Table 1) showed that the TPC values for both purified extracts (MW/US-EtOH 60% after SPE and MW-sbc-H_2_O after SPE) were almost 4 times higher than the values of the crude extracts.

The ANOVA test determined a statistically significant difference between groups for both TPC contents, expressed as GAE mg/g DE or DM (see Section 3.7), which was subsequently checked by Tukey HSD test through multiple comparisons.

In the TPC content expressed as GAE mg/g DE (Table 1), MW/US (g, h), MW-sbc-H_2_O (j) and the corresponding purified samples (by SPE; i and k) resulted to be strongly different from all the other samples. No significant differences were found between TPC content obtained in M40-H_2_O and M75-H_2_O (a and e), M40-H_2_O and M75-EtOH 60% (a and f), M40-EtOH 60% and US-H_2_O (b and c), M40-EtOH 60% and US-EtOH 60% (b and d), M40-EtOH 60% and M75-EtOH 60% (b and f), US-H_2_O and US-EtOH 60% (c and d), US-H_2_O and M75-EtOH 60% (c and f). TPC content in the extract was not significantly influenced by temperature and solvent for maceration and by the last for UAE. 

In the TPC content expressed as GAE mg/g DM (Table 1), samples M40-EtOH 60% and US-H_2_O (b and c), US-EtOH 60% and M75-H_2_O (d and e), in particular, MW/US-EtOH 60% and MW-sbc-H_2_O (h and j), MW/US-EtOH 60% and EM-EtOH 75% (h and l), MW-sbc-H_2_O and EM-EtOH 75% (j and l) were not significantly different. The total yield of phenols from the dry matrix obtained by MW using sbc-H_2_O or combined MW/US with EtOH 60% were comparable between each other and to that obtained by the exhaustive maceration using EtOH 75%, supporting our conclusions.

### 2.4. HPLC Analysis

The identification and quantification required for an analysis of the polyphenol composition of the various extracts was carried out using HPLC. 

Figure 1 shows representative chromatograms of MW-sbc-H_2_O extract before (a) and after purification (b). All the other extracts show a similar profile.

Compounds **1**–**5**, **7** were identified via comparison with the standards chlorogenic acid, *p*-hydroxybenzoic acid, caffeic acid, luteolin-3-glucoside, *p*-coumaric acid and apigenin-3-glucoside. The retention times (RT) and wavelengths of the UV spectra (λ max) of all the identified standards are reported in Table 2.

The retention times (RT), the wavelengths of the UV spectra (λ max), the wavelength used for the quantification (λ for eq.), the equation curves with linearity ranges, R^2^, LOD (limit of detection) and LOQ (limit of quantification) for all the identified standard compounds (ST) are reported in Table 2.

A comparison of the elution orders and UV spectra that are reported in the literature [13,14,15,16,17,18,19,20,21,22,23,24,25,26,27,28,29,30,31,32,33,34,35,36] led to compound 6 being identified as indicated in Table 2.

Peak 6, which was attributed to chicoric acid, was found in US-EtOH 60%, M40-EtOH 60%, MW-sbc-H_2_O, MW/US-EtOH 60% and EM-EtOH 75%.

The quantification of compounds **1**–**5**, **7** was performed using external standards. Quantitative data are expressed as compound amount mg/g DE in Table 3.

It can be seen, in Table 3, that the individual compounds are present in greater quantities in the MW-sbc-H_2_O extract, confirming the TPC value that was obtained using the Folin test (see Section 2.3). 

The extracts MW/US-EtOH 60% and MW-sbc-H_2_O were purified using SPE (see Section 3.5), and analysed by HPLC after SPE. The quantifications of the identified compounds in the purified extracts are also reported in Table 3. A comparison of the values obtained before and after the SPE (see Table 3) showed that the individual compounds for both purified extracts were found in concentrations considerably higher than those obtained in the crude extracts.

Zeb et al. have recently studied the effects of MW cooking on phenolic compounds in chicory leaves [36]. They reported that the amount of *p*-hydroxybenzoic acid obtained was 278 mg/g DE after 20 min of MW irradiation. In our work, however, 427 mg/100 g DE was obtained using MW-sbc-H_2_O. *p*-coumaric acid was found at 33 mg/100 g DE by Zeb et al. [36], while we were able to achieve a higher value of 197 mg/100 g DE. Heimler et al. have reported chlorogenic acid values from 0.21 to 1.02 mg/g DE [23], whereas 3.45 mg/100 g DE was obtained by MW-sbc-H_2_O in our work.

The compounds detected in chicory varieties and reported in the literature [17,37,38,39,40,41,42] were used to confirm our identification results. The compounds, usually reported in the literature but not detected in our extracts, were kaempferol, cyanidin and quercetin derivatives, and acids such as caffeoylquinic acid and caftaric acid. 

### 2.5. Process Design for Scaling Up from Laboratory-Sized Research to Industrial Production

As highlighted in the ‘Polyphenols Market Size and Share-Industry Report 2024’ [43], the worldwide demand for polyphenols will increase from 16,400 tonnes in 2016 (which corresponds to total turnover of 760 million USD) to 34,000 tonnes (1200 million USD) in 2024. Their market is mainly driven by antioxidant products and additives for food, pharma and cosmetics.

This fact combined to the negative cost of chicory leftovers, the use of water as solvent and the consequent reduction of industrial by-products make the process here described attractive for the industry. A midsized industrial plant for a MW-assisted batch treatment of tomato waste was presented by Tabasso et al. in 2019 [44]. The lab-scale aerobic oxidation of biomass was carried out in the same MW professional reactor (SynthWAVE, Milestone), at a temperature of 170 °C for 30 min. For the scale-up, a 1 m^3^ MW reactor with three magnetrons (NL15245, Toshiba) with a thermal insulation, a decanter centrifuge type F2000 for the solid separation (Andritz, Milan, Italy), a nanofiltration system (Evonik) and a spray-drier for the solvent removal were described as main components. Moreover, the energy costs assessment for a biomass batch of 10 kg and 1000 L of solution was calculated, demonstrating that the whole process was profitable from an economic point of view.

We recently showed that cavitational reactors based on high-intensity US and rotational hydrodynamic units can be methods of choice for green extraction within a new technological platform [45]. A full process intensification was guarantee by a pilot scale separation and concentration chain, aiming to create a flow-mode extraction process for the industry. As shown for grape and olive by-products, extraction by means of enabling technologies makes the industrial phenols recovery from agro–food leftovers feasible.

## 3. Materials and Methods 

### 3.1. Materials

Frozen *C. intybus* leftovers were kindly provided by BioSphere (Bertinoro, Italy).

EtOH (ACS grade, ≥99%) (Sigma–Aldrich, Milan, Italy) and methanol (MeOH, HPLC, ≥99.9%) were used for extractions, total phenolic micro-assays and SPE purifications. Acetonitrile for HPLC analyses (MeCN, HPLC Plus, ≥99.9%) was purchased from Sigma–Aldrich, while Milli-Q H_2_O was obtained in the laboratory from a Milli-Q Reference A + System (Merck Millipore, Darmstadt, DE, USA). Glacial acetic acid (AcOH, ≥96%) was purchased from Merck (Darmstadt, DE, USA). Standards of gallic, *p*-hydroxybenzoic, caffeic, ellagic, *p*-coumaric, ferulic, chlorogenic, vanillic, syringic, protocatechuic and 3,4-dimethoxybenzoic (veratric) acids, (−)-epicatechin, apigenin-3-glucoside, luteolin-3-glucoside as well as the Folin–Ciocalteau reagent and sodium carbonate for total phenolic assays were purchased from Sigma-Aldrich (Milan, Italy). 

### 3.2. Water Content Determination

The water content of frozen chicory leftovers was determined using two different methods. Chicory leftovers were freeze-dried for 24 h in a LyoQuest –85 lyophilizer (Telstar, Madrid, Spain) or dried in a muffle furnace at 80 °C for 24 h (Gelman Instrument Company, Ann Arbor, MI, USA)). All analyses were performed in triplicate and expressed as averages ± standard deviation.

### 3.3. Experimental Methods and Reactors

A full plan of experiments was designed with the aim of enhancing the polyphenol content in the crude extracts and these tests were carried out using the extraction conditions described below. Classic extractions (macerations) and non-conventional techniques (UAE, MAE and MAE/UAE) were performed directly on frozen chicory leftovers, using a plant solvent/ratio of 1:15 (*w/v*) and an extraction time of 15 min. Hydroalcoholic mixtures and H_2_O were generally used as the solvents in the single extraction step. In all the experiments, after the reaction, the solid/liquid separation was obtained filtering on a paper filter in a Buchner funnel. When present, EtOH was dried under vacuum at the rotavapor (40 °C), while H_2_O was removed under vacuum in a LyOQuest-85 lyophilizer (Telstar, Madrid, Spain).

#### 3.3.1. Maceration at 40 °C

Conventional maceration of chicory leftovers (20 g) was carried out in a round-bottomed flask in an oil bath at 40 °C under magnetic stirring for 15 min (M40 in the Table 1). H_2_O and EtOH 60% *v/v* were used as the solvents (300 mL).

#### 3.3.2. UAE

UAE was carried out in a 500 mL glass tube, using a probe system (Danacamerini, Turin, Italy) equipped with a titanium horn (ø = 15 mm) with a conical tip (ø = 25mm) for 15 min (US in the Table 1). During the extraction, the temperature was kept near 40 °C by the immersion of the glass tube in an ice bath. The working frequency was 19.5 kHz, while the power was set at 150 W. H_2_O and EtOH 60% *v/v* were used as the solvents (300 mL) in a single extraction step (20 g of frozen chicory leftovers).

#### 3.3.3. Maceration at 75 °C

Conventional maceration of frozen chicory leftovers (10 g) was carried out in a round-bottomed flask in an oil bath at 75 °C under magnetic stirring for 15 min (M75 in the Table 1). H_2_O and EtOH 60% *v/v* were used as the solvents (150 mL). 

#### 3.3.4. Combined MW/US Procedure

The combined MW/US extraction process was performed in a glass pear-shaped bottomed-flask (250 mL) placed in a multimode MW reactor (MicroSYNTH, Milestone, Bergamo, Italy; working frequency 2.45 GHz, maximum power 1500 W), which was combined with a Pyrex^®^ horn, used as the US source (working frequency 20.3 kHz, maximum power 60 W) (MW/US in the Table 1). The temperature, which was controlled by the MW optical-fibre thermometer inside the glass vessel, was increased from 10 °C (10 g of frozen chicory leftovers in 150 mL of solvent) to 75 °C in 5 min and then kept at 75 °C for 15 min by the simultaneous MW/US irradiation. To better control the temperature rise, the extraction flask was immersed in a bath, connected to an efficient cooling system consisting of a chiller that refrigerates and circulates a MW-inert liquid (Galden H270, Solvay-Solexis, Milan, Italy). The MW and US power were set at 400 W (150 W mean power used) and 30 W, respectively. H_2_O and EtOH 60% *v/v* were used as the solvents in a single extraction step. 

#### 3.3.5. MAE Under Pressure

MAE was performed in a Teflon® tube (1 litre) directly inserted in a closed multimode reactor (SynthWAVE, Milestone, Bergamo, Italy; working frequency 2.45 GHz, maximum power 1500 W) under magnetic stirring and N_2_ pressure (20 bar) (MW-sbc-H_2_O in the Table 1). H_2_O was used as solvent under sub-critical conditions. The temperature, which was controlled by a thermocouple inside the vessel, was increased from 10 °C (20 g of frozen chicory leftovers in 300 mL of H_2_O) to 150 °C in 5 min (set power 1500 W, mean power used 1200 W) and then kept at 150 °C for 15 min (set power 800 W, mean power used 500 W). 

#### 3.3.6. Exhaustive Protocol

The exhaustive extraction of frozen chicory leftovers (10 g) was carried out in a round-bottomed flask (250 mL) at reflux at 85 °C in an oil bath under magnetic stirring, using EtOH 75% *v/v* as the solvent (150 mL) for 240 min (EM in the Table 1). 

### 3.4. Total Phenolic Assay

TPC was determined according to the method developed by Cicco et al. for the crude extracts of the traditional and non-conventional procedures [46]. Quantification was carried out according to a standard curve (curve equation: y = 0,0016x − 0,0086; R² = 0,9996) and using appropriate dilutions of a solution of gallic acid (between 5 and 160 mg/L) in a H_2_O/MeOH 8:2 mixture as the reference phenolic compound. Extract solutions were prepared at a concentration of 1 mg/mL in a H_2_O/MeOH 8:2 mixture. The gallic acid and extract solutions (250 µL) were placed into test tubes. The following solutions were added sequentially to each tube: 250 µL of Folin–Ciocalteu (diluted 1:1 with distilled H_2_O), 500 µL of 10% *p/v* Na_2_CO_3_ solution, 4 mL of distilled H_2_O. The resulting solution was vigorously shaken and left at room temperature for 25 min prior to analysis. The absorption of the final mixtures was measured at 740 nm, in a 1 cm cuvette, using a Cary 60 UV-Vis spectrophotometer (Agilent Technologies, Santa Clara, CA, USA). These conditions provided the assay with high accuracy and reproducibility. TPC was expressed as gallic acid equivalents (GAE, mg/g) over the dried extract (DE) and gallic acid equivalents (GAE, mg/g) over the dried matrix (DM). All analyses were performed in triplicate and expressed as averages ± standard deviation.

### 3.5. Extract Purification

In order to obtain extracts that were richer in the polyphenolic fraction, some extracts were purified using SPE on a C18 Sep-Pak cartridge (Waters). 50 mg of sample were dissolved in 1 mL 0.5% AcOH. The bonded phase was solvated three times using 2 mL of pure MeOH and three times with 2 mL of 0.5% AcOH. The sample solution was then loaded onto the cartridge. The unwanted components (fraction 1) were successively eluted using AcOH 0.5% (three times, 2 mL) as the solvent. The second fraction, including all of the components of interest, was eluted with 80% MeOH (three times, 2 mL). Finally, the used cartridge was discarded with pure MeOH (three times, 2 mL).

### 3.6. HPLC Analyses

HPLC analyses were performed on a Waters binary pump 1525 linked to a 2998 PDA (Waters Corp., Milford, CT, USA), using a Synergi Hydro RP C18 column (250 mm, 4.6 mm, 5 μm; Phenomenex, Torrance, CA, USA) and 2% AcOH (A) and MeCN (B) as the mobile phases. The monitored wavelengths were 280 and 340, while three-dimensional data were acquired in the 200–600 nm range. The gradient program started from 0% B, which was maintained for 6.5 min, up to 50% B over the 6.5–30 min period, from 50% to 100% B over 30–36 min, followed by a 100% B step at 36–42 min. 

The quantification of polyphenols compounds was performed using calibration curves obtained with external standards. Standard solutions of chlorogenic acid, *p*-hydroxybenzoic acid, caffeic acid, luteolin-3-glucoside, *p*-coumaric acid and apigenin-3-glucoside were analyzed by HPLC (20 μL injection) to give linear regressions with R^2^ > 0.999. Equation curves, the wavelength used for the quantification, R^2^, linearity range, LOD and LOQ for each standard analysed are indicated in the Table 2. All samples were dissolved in H_2_O/MeOH 8:2 before injection, giving concentrations of between 5 and 10 mg/mL. 

### 3.7. Statistical Analyses

Analyses were performed all in triplicate. Differences between the different samples means were evaluated by one-way analysis of variance (ANOVA) and then with the Tukey’s HSD multiple comparison test. ANOVA test on extraction yields gave F = 917.52, *p* value = 3.68 × 10^−24^, F_crit_ = 2.39, on TPC content GAE mg/g DE, F = 13971.05, *p* value = 7.17 × 10^−43^, F_crit_ = 2.21, while on TPC content GAE mg/g DM, F = 3902.15, *p* value = 1.95 × 10^−30^, F_crit_ = 2.39.

For Tukey HSD test, an α= 0.05 value was used. For extraction yields, considering k = 10, n = 3, N = 30, MSW = 1.366, q = 5.008, an HSD = 3.38 value was obtained. For TPC content (GAE mg/g DE), starting from k = 12, n = 3, N = 36, MSW = 1.080, q = 5.099, the calculated HSD value was 3.059, while for TPC content (GAE mg/g DM), starting from k = 10, n = 3, N = 30, MSW = 0.149, q = 5.008, the HSD value was 1.117. The differences between the samples means were considered significant when higher than HSD value.

## 4. Conclusions

In conclusion, this work highlights the possibility of efficiently extracting the phenolic fraction from chicory leftovers under UAE, MAE and MW/US irradiation. These techniques are very fast and efficient procedures and their higher extraction selectivity strongly improves polyphenol extraction from chicory, as compared to traditional methods; process time, the solvent amount and energy consumption are all reduced. 

The best results, in terms of extractive phenol yields, were achieved after only 15 min of MAE with sub-critical H_2_O and using combined MW/US with a 60% *v/v* EtOH solution.

The results of our work may become the basis for the future development of the MAE and UAE methods, which are known to be extremely versatile and suitable for easy scale-up design [44,45]. 

## Figures and Tables

**Figure 1 molecules-24-02681-f001:**
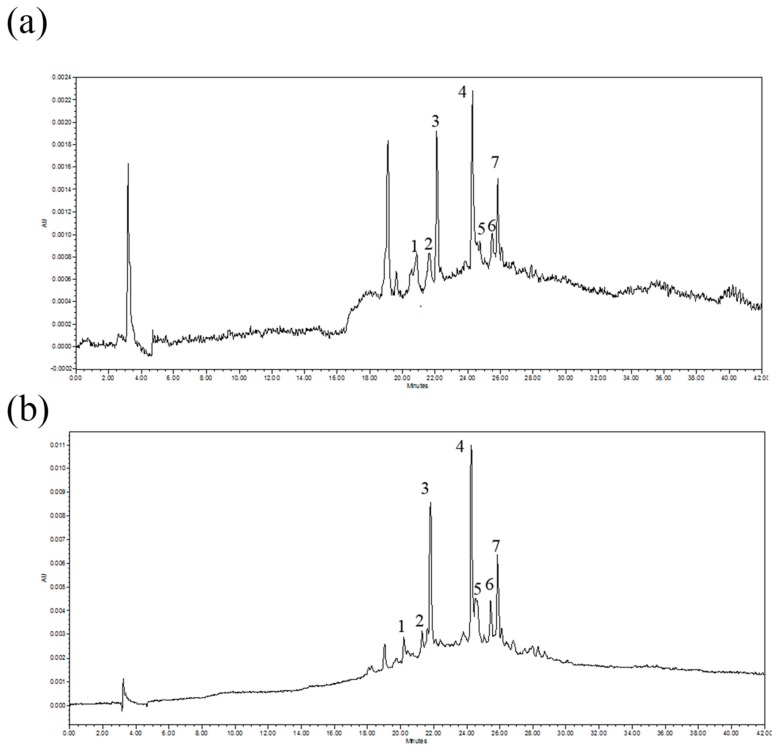
HPLC profiles at λ = 340 nm of the extract obtained using microwave (MW) and sub-critical H_2_O (**a**) before and (**b**) after solid phase extraction (SPE). Peaks on the chromatogram correspond to: 1, chlorogenic acid; 2, *p*-hydroxybenzoic acid; 3, caffeic acid; 4, luteolin-3-glucoside; 5, *p*-coumaric acid; 6, chicoric acid; 7, apigenin-3-glucoside.

**Table 1 molecules-24-02681-t001:** Extraction yields expressed as a *w/w* percentage over dried matrix (DM); total phenolic content, expressed as GAE mg/g DE, GAE mg/g DM and GAE g/kg FM, obtained using different extraction conditions.

Extraction Method	Sample ^a–l^	Temp. (°C)	Time (min)	Yield (% *w/w*)	TPC (GAE mg/g DE)	TPC (GAE mg/g DM)	TPC (GAE g/kg FM)
Maceration	M40-H_2_O ^a^	40	15	33.9 ± 0.53	30.4 ± 1.13 ^ef^	10.3 ± 0.12	0.72
	M40-EtOH 60% ^b^			44.6 ± 0.46 ^c^	34.6 ± 0.80 ^cdf^	15.4 ± 0.37 ^c^	1.08
UAE	US-H_2_O ^c^	40	15	42.8 ± 1.15 ^be^	35.2 ± 0.61 ^bdf^	15.0 ± 0.26 ^b^	1.05
	US-EtOH 60% ^d^			50.5 ± 1.36	37.0 ± 0.28 ^bc^	18.7 ± 0.15 ^e^	1.31
Maceration	M75-H_2_O ^e^	75	15	40.5 ± 0.48 ^c^	29.3 ± 0.36 ^a^	11.9 ± 0.08 ^d^	0.83
	M75-EtOH 60% ^f^			57.1 ± 0.49	32.5 ± 1.16 ^abc^	18.5 ± 0.37	1.30
MW/US	MW/US-H_2_O ^g^	75	15	66.9 ± 0.85 ^j^	41.7 ± 0.37	27.9 ± 0.24	1.95
	MW/US-EtOH 60% ^h^			87.0 ± 1.48	49.7 ± 0.44	43.3 ± 0.29 ^jl^	3.03
	MW/US-EtOH 60%after SPE ^i^	x	x	5.0% *	168.8 ± 1.06	x	x
MAE	MW-sbc-H_2_O ^j^	150	15	65.4 ± 0.79 ^g^	67.5 ± 1.17	44.2 ± 0.65 ^hl^	3.09
	MW-sbc-H_2_O after SPE ^k^	x	x	11.6% *	258.6 ± 1.16	x	x
Exhaustivemethod	EM-EtOH 75% ^l^	85	15	95.9 ± 1.19	46.1 ± 0.79	44.2 ± 0.18 ^hj^	3.09

* Solid phase extraction (SPE) purification yields, excluded from Tukey post-hoc test. ^a^^–l^ These letters refer to the treatment indicated in the Table rows in the Sample column; in the yield and total phenolic content (TPC) (GAE eq. mg/g DE or DM) columns, the letters indicate the corresponding treatments which are not significantly different with the one present in the row (alpha = 0.05, Tukey HSD (honestly significant difference) post-hoc test).

**Table 2 molecules-24-02681-t002:** Retention times (RTs), λ max, λ for eq., equation curves, linearity ranges, R^2^, LOD (limit of detection) and LOQ (limit of quantification) for all the identified standard compounds (ST).

Peak	Identif.	Compound	RT (min)	λ max (nm)	λ for Eq. (nm)	Equation Curve (mg/mL)	Lin. Range (mg/mL)	R^2^	LOD (mg/mL)	LOQ (mg/mL)
1	ST	Chlorogenic acid	20.0	215, 240, 326	340	y = 2,078,561.4x − 652.1	0.003–0.200	0.9998	0.001	0.003
2	ST	*p*-Hydroxybenzoic acid	21.0	256	280	y = 1,001,749.2x + 1456.8	0.004–1.27	1.0000	0.001	0.004
3	ST	Caffeic acid	21.8	217, 240, 298, 324	340	y = 4,699,291.5x − 3720.2	0.002–0.200	0.9999	0.001	0.002
4	ST	Luteolin-3-glucoside	23.9	203, 254, 348	340	y = 3,376,292.6x − 5788.6	0.004–0.200	0.9999	0.002	0.004
5	ST	*p*-Coumaric acid	24.7	217, 235, 323	280	y = 4,304,744.2x − 1271.3	0.0015–0.160	0.9999	0.0007	0.0015
6	R *	Chicoric acid	25.4	241, 305, 327	-	-	-	-	-	-
7	ST	Apigenin-3-glucoside	25.8	266, 308, 337.6	340	y = = 1,870,507.7x − 434.2	0.003–0.200	0.9999	0.001	0.003

* R: reference from the literature.

**Table 3 molecules-24-02681-t003:** Quantification of the identified compounds. Data are expressed as mg/g dried extract (DE).

Sample	Chlorogenic Acid	*p*-Hydroxy-benzoic Acid	Caffeic Acid	Luteolin-3-glucoside	*p*-Coumaric Acid	Apigenin-3-glucoside
M40-H_2_O	* <LOQ	* <LOQ	2.83	1.19	0.69	1.06
M40-EtOH 60%	* <LOQ	2.55	1.86	1.39	** N.D.	0.90
US-H_2_O	* <LOQ	2.63	1.96	1.33	** N.D.	0.73
US-EtOH 60%	* <LOQ	2.72	1.74	1.06	** N.D.	0.72
M75-H_2_O	3.52	1.99	2.45	1.68	0.64	1.49
M75-EtOH 60%	1.92	2.67	1.94	1.63	** N.D.	1.16
MW/US-H_2_O	0.82	2.84	1.76	1.32	** N.D.	2.76
MW/US-EtOH 60%	1.10	3.14	2.86	2.42	1.30	1.79
MW/US-EtOH 60% after SPE	9.75	6.44	6.17	5.56	4.02	6.27
MW-sbc-H_2_O	3.45	4.27	3.16	3.57	1.97	5.09
MW-sbc-H_2_O after SPE	12.2	9.35	8.75	7.06	4.36	15.8
EM-EtOH 75%	1.70	2.35	2.59	1.80	0.57	4.47

* values lower than LOQ were detected for these analysis; ** N.D.: not detected.

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
