# Peer review of "Technology and Process Design for Phenols Recovery from Industrial Chicory (*Chicorium intybus*) Leftovers"

_molecules, 2019, doi:10.3390/molecules24152681_

Round 1
Reviewer 1 Report
This paper investigated the influence of ultrasound-assisted extraction (UAE), microwave-assisted extraction (MAE) and their combination on the rapid and selective recovery of the phenolics from chicory leftovers.
The article is well written, and of interest to the scientific community because is designing a green and efficient extraction process of phenolic compounds from chicory as valid alternatives to traditional methods. As compared to traditional methods, these techniques were very fast and efficiently, and their higher extraction selectivity strongly improved polyphenol extraction from chicory; process time, the solvent amount and energy consumption were reduced.
The methods used are well described and it is appropriate. In terms of extractive phenol yields, the best results were achieved after 15 min of MAE and using a combination of MAE/UAE with a 60% (v/v) aqueous ethanol. The results may become the basis for the future development of the MAE and UAE methods, which are suitable for easy scale-up design.
However, this work has some flaws that need to be addressed before publication. I believe that this work will be better with suggested revisions.
Introduction
The Introduction could be shortened and moved into the Discussion.
Results and discussion
– Pg.3, L106: The sentence “Water content of frozen… and 92.9±0.17 weight/weight percentage…” should be replaced as “Water content of frozen… and 92.9±0.17 weight/weight (w/w) percentage…”
2.2. Extraction conditions
– The frozen chicory was used for extraction. As presented in Table 2, total phenolic content was expressed as GAE mg/g dry extract, GAE mg/g dry matter, and GAE g/kg fresh matter. The fresh chicory did not describe as material for extraction.
2.4. Phenolic content
– TPC (GAE mg/g DE) and TPC (GAE mg/g DM) in Figure 1 and Table 2 are the same values. I suggest the reduction of presented data based on results shown in Table 1 and 2 and Figure 1. Eg. time and temperature are the same in Table 1 and 2, and TPC (GAE mg/g DE) and TPC (GAE mg/g DM) in Figure 1 and Table 2 are the same values. Did the TPC (GAE g / kg FM) redundant since fresh material is not included in the description of materials which were extracted?
2.5. HPLC analysis
– The qualitative analysis of phenolic compounds in extracts was obtained by comparison with the six standards. Is there a special reason why the chicoric acid is not identified in the same way as previously six?
– In addition, quantitative analysis of these six was performed using external standards and expressed as amount mg/g DE. Please, describe the linearity range for each compound and LOD and LOQ.
2.6 Extract purification
– MW-sbc-H2O and 242 MW/US-EtOH 60% extracts were purified using SPE because they showed the highest TPC values. I suggest that the results in Table 5 could be added in Table 2 (or in Figure 1). On this way, improving in the TPC (GAE mg/g DE) after SPE purification will be easier to compare.
– In the same reason, the results shown in Table 6 could be moved in Table 4.
– Next, I suggest the addition of the chromatogram of purified extract in Fig.2.
Materials and methods
– Pg. 9, L290: Is chicory leftovers fresh or frozen?
3.4. Total phenolic micro-assay
– Why micro-assay? Total volume is 5 ml. This method is not on the microplate.
– Quantification of TPC was carried out according to a standard curve of gallic acid (concentration range between 5 and 160 mg/L). Add equation of standard curve and R square (R2).
Additionally:
The Graphical abstract is missing.
Conclusion:
This manuscript can be accepted after the implementation of suggested changes and corrections.
Author Response
The manuscript was thoroughly revised highlighting all the changes. We acknowledge the reviewers for the useful amendments.

Reviewer 2 Report
The manuscript shows a good and proper work about the application of some green and efficient extraction methods to isolate polyphenols from chicory. Ultrasound-assisted extraction, microwave-assisted extraction and their simultaneous combination were compared between them and with a classical method, maceration, using different extraction conditions (solvent, temperature, time). The methods are clear and well described, the results are relevant and well presented, the conclusion is appropriate.
I recommend the acceptation of the manuscript to be published with minor corrections:
Page 3, lines 100-103: These two sentences show some results of the work and I suggest to move them in the section 2 (Results and discussion).
Page 4, line 172: I suggest to write ”selected” instead to ”created”.
Page 6, line 210: ”profile” instead to ”profiles”
Page 7, line 228: I suggest to write ”individual” instead to ”single”.
Page 9, line 319: Lined-up.
Author Response

(The authors gave the same response as above.)

Reviewer 3 Report
The work fits in the line of similar studies already published for the extraction of polyphenol in industrial vegetal leftovers using UAE, MAE or a combination as well as US irradiation. Nevertheless, scarce information has been published around chicory leftovers. Therefore, I consider that this document presents a certain novelty.
As the chromatographic results include the presence of phenolic acids, the word polyphenols in the title must be changed to phenols.
In table 1 standard deviation of Yield must be added and it will be necessary to indicate the number of extractions performed for each treatment to obtain significant values.
In figure 1 the specification of DE and DM in the figure caption must be added, a test for significant differences such as Tuckey test must be added to a better data analysis
A better resolution of figure 2, with a description of the components in the figure, could have resulted in a better presentation.
As reported by the authors after purification an increment of the TPC is logically obtained, but the interesting thing in this part could be the presence and increment of the biological activity
The description of the phenol quantification methodology by HPLC must be indicated in the methodology section
As suggested by the authors the aim of the work was the investigation of the influence of UAE, MAE and the combination. Nevertheless, information related to more than the reduction in time and the increment of recovery must be added, i.e. use of energy, evaluation of biological activity….
A higher phenolic extraction yield is obtained, but are these compounds biologically actives using these techniques?, I strongly recommend the determination of a biological activity
The title suggests the presence of a technology and process design … but in the text is not clearly indicated
Author Response

(The authors gave the same response as above.)

Round 2
Reviewer 1 Report
The manuscript " Technology and process design for phenols recovery from industrial chicory (Chicorium intybus) leftovers" by Cravotto et al., was significantly improved compared to the previously submitted version. The authors responded to all the suggestions of the reviewer and accordingly added all changes in the text.
The number of Tables and Figures was reduced, and the data are noticeable and comparable. In the new Figure 1, the comparison of HPLC chromatograms before and after SPE was shown and qualitative and quantification parameters were presented in the amended Table 2.
In addition, statistical analysis was added (ANOVA test for analyses of variance followed by the Tukey HSD post-hoc test of multiple 23 comparisons). The discussion was extended and supplemented with required parts (e.g. statistical analysis, process design for scaling up from laboratory-sized research to industrial production), and references were adequately set in accordance with changes made.
Finally, this work is now significantly improved.
Reviewer 3 Report
Authors have taken into consideration the comments performed to improve the quality of the document